# The Crystal Structure of the Hsp90-LA1011 Complex and the Mechanism by Which LA1011 May Improve the Prognosis of Alzheimer’s Disease

**DOI:** 10.3390/biom13071051

**Published:** 2023-06-28

**Authors:** S. Mark Roe, Zsolt Török, Andrew McGown, Ibolya Horváth, John Spencer, Tamás Pázmány, László Vigh, Chrisostomos Prodromou

**Affiliations:** 1Department of Biochemistry and Biomedicine, University of Sussex, Brighton BN1 9QG, UK; m.roe@sussex.ac.uk; 2Institute of Biochemistry, Biological Research Centre, 6726 Szeged, Hungary; Torok.zsolt@brc.hu (Z.T.); hibi@lipidart.com (I.H.); Vigh.laszlo@brc.hu (L.V.); 3Sussex Drug Discovery Centre, School of Life Sciences, University of Sussex, Brighton BN1 9QG, UK; a.mcgown@sussex.ac.uk (A.M.); j.spencer@sussex.ac.uk (J.S.); 4Gedeon Richter Plc, 1475 Budapest, Hungary; tpazmany@yahoo.com; 5National Vaccine Factory Plc, 4032 Debrecen, Hungary

**Keywords:** Alzheimer’s disease, Hsp90, FKBP51, immunopilin, LA1011, co-chaperone, tau, drug discovery

## Abstract

Functional changes in chaperone systems play a major role in the decline of cognition and contribute to neurological pathologies, such as Alzheimer’s disease (AD). While such a decline may occur naturally with age or with stress or trauma, the mechanisms involved have remained elusive. The current models suggest that amyloid-β (Aβ) plaque formation leads to the hyperphosphorylation of tau by a Hsp90-dependent process that triggers tau neurofibrillary tangle formation and neurotoxicity. Several co-chaperones of Hsp90 can influence the phosphorylation of tau, including FKBP51, FKBP52 and PP5. In particular, elevated levels of FKBP51 occur with age and stress and are further elevated in AD. Recently, the dihydropyridine LA1011 was shown to reduce tau pathology and amyloid plaque formation in transgenic AD mice, probably through its interaction with Hsp90, although the precise mode of action is currently unknown. Here, we present a co-crystal structure of LA1011 in complex with a fragment of Hsp90. We show that LA1011 can disrupt the binding of FKBP51, which might help to rebalance the Hsp90-FKBP51 chaperone machinery and provide a favourable prognosis towards AD. However, without direct evidence, we cannot completely rule out effects on other Hsp90-co-chaprone complexes and the mechanisms they are involved in, including effects on Hsp90 client proteins. Nonetheless, it is highly significant that LA1011 showed promise in our previous AD mouse models, as AD is generally a disease affecting older patients, where slowing of disease progression could result in AD no longer being life limiting. The clinical value of LA1011 and its possible derivatives thereof remains to be seen.

## 1. Introduction

Heat shock protein 90 (Hsp90) is a molecular chaperone that together with its co-chaperones is responsible for the activation of an eclectic set of client proteins [1]. These clients represent key signalling proteins, which when dysregulated lead to disease, including cancer and neurodegeneration [2,3,4,5]. The role of Hsp90 is now being recognized in Alzheimer’s disease (AD) [6,7,8,9], which is the most common neurodegenerative disease in humans and is rising due to increasing longevity. It appears that the Hsp90 interaction with tubulin-associated unit (tau) promotes a self-aggregating open tau conformation that was identified in the formation of small tau oligomers [9]. Evidence suggests that the Hsp90 interaction exposes the aggregation-prone repeat domain to other tau molecules, but within the context of Hsp90, the formation of tau fibrils is inhibited or blocked. Thus, dysregulation of this complex has the potential to lead to tau aggregation. It may also be significant that the levels of all Hsp90 paralogs reduce in regions of the brain affected by AD, whereas tau levels remain unaltered [7]. The structure of Hsp90 in complex with Tau has been previously determined [8].

It has been proposed that the overexpression of mutant forms of β-amyloid precursor protein (APP) leads to amyloid-β (Aβ) plaque and tau neurofibrillary tangle formation [5]. Hsp90 forms macromolecular complexes with co-chaperones that can regulate tau metabolism and Aβ processing [10]. The current model suggests that Aβ peptide triggers the Hsp90-directed hyperphosphorylation of tau [11,12], which subsequently leads to neurofibrillary tangles and neurotoxicity [13,14,15,16,17,18,19]. However, despite significant amounts of research, further clarification of the pathogenetic mechanism is needed.

Abnormal or modified tau can trigger its own degradation via the recruitment of carboxy-terminus of Hsc70-interacting protein (CHIP), an E3 ligase co-chaperone of Hsp90. CHIP drives ubiquitination and subsequently promotes the downstream proteasomal degradation of tau [20,21]. However, Hsp90 also associates with a variety of other co-chaperones, including the peptidylprolyl cis/trans isomerase immunophilins FK-506-binding protein 51 (FKBP51) and FKBP52 and cyclophilin 40 (Cyp40, also known as peptidylprolyl isomerase D) [22]. The latter was recently shown to be neuroprotective as it is able to disaggregate tau fibrils in vitro and, significantly, it prevents toxic tau accumulation in vivo [23]. However, Cyp40 appears to decrease with age and is also repressed in AD [24]. In contrast, FKBP51 is able to preserve toxic tau oligomers in vivo [25], and mice lacking FKBP51 have been shown to have decreased tau levels in the brain [25,26]. Significantly, FKBP51 levels appear to progressively increase with age and further increase in AD patients [25,27]. Stress is also known to increase FKBP51 and is associated with AD [28,29]. The structure of FKBP51 in complex with Hsp90 was recently reported, and it shows that the seventh helix of the TPR domain of FKBP51 binds across a hydrophobic cleft or pocket formed at the dimer interface at the MEEVD peptide end of the C-terminal domain of Hsp90 [30]. The interaction is critical for aligning the active PPIase domain of FKBP51 with the bound tau client.

In contrast, FKBP52, by a direct molecular interaction with truncated and wild-type tau, induces its aggregation [31,32]. However, FKBP52 levels appear to be lower in the cortex of AD patients’ brains [24,33]. Another co-chaperone that plays a role in the phosphorylation state of tau is protein phosphatase 5 (PP5). The latter is a Ser/Thr phosphatase that is activated when bound to Hsp90 [34] and is able to dephosphorylate tau at several phosphorylation sites connected to AD pathology [35]. Significantly, PP5 activity is repressed in AD [36]. While the list of co-chaperones that interact with Hsp90 to affect the phosphorylated state of tau may be more extensive, FKBP51, FKBP52, PP5 and CHIP appear to be major players, and consequently, imbalances in these co-chaperones could have major implications in the development of AD [22]. Clearly, decreases in FKBP52 and PP5 in the AD brain, with co-current increases in FKBP51, would favour the hyperphosphorylation of tau. Thus, the maintenance of healthy homeostasis represents a prime focus and is a key therapeutic target for drug discovery for AD and neurodegenerative diseases [37,38]. Consequently, there has been a drive to develop Hsp90 inhibitors that promote the degradation of tau [10,39,40] or alternatively to identify small molecules that can restore the normal balance of chaperone and co-chaperone systems that decline with age or are abnormally altered through stress.

Previously, we found that the dihydropyridine LA1011 could increase dendritic spine density and reduce tau pathology and amyloid plaque formation in transgenic AD mice [41]. Further work identified the C-terminal domain of Hsp90 as the target for LA1011 binding, and molecular dynamics suggested a specific site to which it bound. Attempts to confirm the site were not wholly conclusive, because mutagenesis of the proposed binding site did not provide a means to prevent binding of LA1011 completely [42]. Nonetheless, these studies have identified LA1011 as an inducer of Hsp90 ATPase activity and the heat shock response and as an effective small molecule against AD in mouse models. Here, we report the crystal structure of a C-terminal domain of Hsp90 in complex with LA1011 and suggest that its binding may help to restore normal levels of the FKBP51–Hsp90 complex, a main player in tau hyperphosphorylation. However, without direct evidence of this, it is possible that LA1011 may also mediate its action by disrupting other immunophilin–Hsp90 complexes or even Hsp90 client proteins, which may be contributing to a change in tau phosphorylation or its ability to form aggregates. Whatever the precise mechanism, the clinical evaluation of LA1011 is urgent because small effects that can slow the progression of the disease in elderly patients could have significant benefits to a point that AD itself would no longer be life limiting.

## 2. Methods and Materials

### 2.1. Protein Purification

The yeast Hsp90 C-terminal domain (amino acid residues 438 to 677) was cloned into p3E (A. Oliver, University of Sussex) and expressed in *Escherichia coli* BL21(DE3) as a GST-tagged fusion. FKBP51 and FKBP51-Δ7He (lacking residues 401–457 of the extended 7th helix of the TPR domain) were a kind gift from D. Southworth (UCSF Weill Institute for Neurosciences) and were expressed with an N-terminal His-tag [30]. The Hsp90 fusion was initially purified using GSH resin and then cleaved with PreScission protease, as previously described [43]. In contrast, FKBP51 constructs were initially purified using Talon affinity metal chromatography [44]. All proteins were further purified using Superdex 200 HR gel filtration and Q-Sepharose ion-exchange chromatography, as previously described [44]. Proteins were dialysed against 20 mM Tris pH 7.4 containing 5 mM NaCl and 1 mM EDTA.

### 2.2. Crystallization and Refinement

The Hsp90-LA1011 complex was crystallized using the sitting-drop method with protein at 22.5 mg mL^−1^ in 20 mM Tris pH 7.4 containing 5 mM NaCl and 1 mM EDTA and 5 mM LA1011 against wells containing 100 mM MES pH 7.5, 200 mM magnesium chloride hexahydrate, 15% PEG Smear Medium (12.5% *w*/*v* of each: PEG 3350, 4000, 2000 and 5000 MME) and 5% (*v*/*v*) 2-propanol at 14 °C (H3 from the BCS Screen HT-96, Molecular Dimensions, MD1-105). Crystals were harvested by successive transfer into a crystallization buffer with increasing ethylene glycol content to 30% and then flash-cooled in liquid nitrogen. Diffraction data were collected from crystals cooled to 100 K on I24 at Diamond Light Source. Data were automatically processed with autoPROC/STARANISO (STARANISO available at http://staraniso.globalphasing.org/cgi-bin/staraniso.cgi, Cambridge, UK: Global Phasing Ltd.; run through the Diamond iSpyB pipeline on 19 February 2020.) [45], and asymmetric unit contents were estimated using the Matthews coefficient (*CCP*4 suite) [46]. The LA1011–Hsp90 structure was solved using *Phaser* (*CCP*4) with a model created from PDB 2CGE. The structure was refined with *REFMAC* 5.7 [47], and manual rebuilding was performed in *Coot* [48]. The structure was deposited in the PDB database (PDB 8OXU) and was displayed using *PyMOL* (Schrödinger, L. & DeLano, W., 2020. *PyMOL*, Available at: http://www.pymol.org/pymol).

### 2.3. Isothermal Titration Calorimetry

The heat of interaction was measured on an ITC_200_ microcalorimeter (Microcal) under the same buffer conditions (20 mM Tris, pH 7.5, containing 5 mM NaCl). Aliquots of 1 mM LA1011 were injected into 30 μM of yeast Hsp90 or Hsp90-FKBP51 complex (consisting of 30 μM Hsp90 and 60 μM FKBP51 or FKBP51-Δ7He) at 30 °C. Interactions with FKBP51 and FKBP51-Δ7He were performed by injecting aliquots of 400 μM of intact FKBP51 or FKBP51-Δ7He into 30 μM Hsp90 or 30 μM Hsp90 in complex with 1 mM LA1011. Heats of dilution were determined by diluting the injectant into the buffer. Data were fitted using a curve-fitting algorithm (Microcal Origin).

## 3. Results and Conclusions

### 3.1. The Crystal Structure of Hsp90-LA1011

The structure of a complex between LA1011 (Figure 1A) and a Hsp90 fragment (residues 438–677) was solved with molecular replacement and refined to an anisotropic resolution limit of 2.94 A (isotropic 3.18 Å; Table 1). The structure revealed two dimer molecules of the Hsp90 fragment in the asymmetric unit, but only one of the dimers contained bound LA1011. Superimposition of the LA1011-bound Hsp90 dimer with the apo-dimer revealed that the unoccupied site is restricted (especially by the side chain of Leu 674) and requires movement of a helix (Pro 661 to Leu 676) to form an open site able to accept LA1011 (Figure 1B). It is likely that a crystal where both sites are occupied with LA1011 may not be readily crystallizable, as we did not observe such crystals. However, LA1011 was found to bind in a small hydrophobic pocket at the extreme C-terminus of the dimer interface of Hsp90 (Figure 1C–E). This is in contrast to and at the opposite end of the C-terminal domain of the previous binding prediction using molecular dynamics [42]. However, one of the methyl ester groups of LA1011 bound across the two-fold symmetry point of the Hsp90 dimer, and hence, only a single molecule of LA1011 was able to bind to the Hsp90 dimer (Figure 1F). This is in agreement with previous binding studies using isothermal titration calorimetry (ITC) [42]. The binding pocket was lined by a series of hydrophobic and hydrophilic amino acid residues, but there were no obvious hydrophilic interactions (Figure 1D,E). The pocket was lined by the amino acids Leu 671, Ile 672, Leu 674, Gly 675 and Leu 676 from one molecule of the dimer and by Glu 624, Leu 625, Arg 628, Thr 638, Asp 641, Leu 642, Leu 671 and Leu 674 from the other protomer.

### 3.2. LA1011 and FKBP51 Competitively Bind to Hsp90

The structure of Hsp90 in complex with FKBP51 has previously been determined [30], which shows that helix 7 of the TPR domain of FKBP51 is docked in a pocket or cleft formed at the extreme C-terminus of the Hsp90 dimer interface (Figure 2). Previously, molecular dynamics predicted that LA1011 binds to a similar pocket at the other end of the C-terminal dimer interface [42]. However, mutagenesis could not prove beyond any doubt that this site was the actual binding site observed in interaction studies, because there was a lack of residues that could be mutated to sterically hinder binding altogether. Consequently, we wondered whether the true binding site could be the same as that seen for helix 7 of the TPR domain of FKBP51. We therefore decided to use isothermal titration calorimetry (ITC) to investigate whether FKBP51 binding could block LA1011′s interaction with Hsp90 and vice versa. The numerical data for ITC experiments hereafter, which are shown in Figure 3, are summarized in Appendix A.

Since only one molecule of FKBP51 is able to interact with the extreme C-terminal hydrophobic pocket of Hsp90 but two Hsp90 MEEVD peptide-binding sites are available, we expected that two different binding events should be detectable with ITC for intact FKBP51, but a single site would be a better fit for the FKBP51 mutant lacking the helical extension (amino acid residues 401–457; FKBP51-Δ7He) of the TPR domain. As expected, full-length FKBP51 bound to Hsp90, and we found that the one-site model fitting of the data was poor compared to a two-site model fit (Figure 3A,B). The *K*d values for the two-site binding were 0.5 and 32.4 μM (Figure 3B), indicating that the C-terminal extension of the TPR domain of FKBP51 is an essential component for high-affinity binding to Hsp90. We next measured the affinity of binding for FKBP51-Δ7He with Hsp90, which we expected would produce data that would conform to a one-site model fit, since two molecules of the construct would have equivalent binding sites (MEEVD only). As expected, using a one-site model fit, we found that this construct bound with a *K*d of 11.6 μM and with a stoichiometry that was close to 1 (1:1.25, Hsp90:FKBP51-Δ7He; Figure 3C). The affinity for this interaction was similar to the weaker binding affinity obtained using the two-site model fit with intact FBBP51, which was due to MEEVD binding alone (32.4 μM; Figure 3B). This indicates, as we had expected, that high-affinity binding to Hsp90 had been compromised with the FKBP51-Δ7He mutant. We next measured the binding affinity of our most recent sample of LA1011 and found it to bind to Hsp90 with a *K*d of 13.1 μM (Figure 3D), which was similar to the measurement seen in our previous studies (*K*d = 13.5 μM) [42]. To investigate the binding of LA1011 to an Hsp90-FKBP51 complex, we used a 30 μM Hsp90 solution saturated with 60 μM of intact FKBP51. We found that the binding of LA1011 to the Hsp90-FKBP51 complex was compromised (*K*d = 108 μM; Figure 3E), but it was restored when using Hsp90 saturated with the FKBP51-Δ7He mutant (*K*d = 17.9 μM; Figure 3F). The affinity for the binding of LA1011 to Hsp90-FKBP51-Δ7He was similar to that seen for the binding of LA1011 to free Hsp90 (*K*d = 13.1 μM; Figure 3D). The results so far suggest that the helical extension of the TPR domain of FKBP51 and LA1011 compete for binding to the extreme C-terminal hydrophobic pocket of Hsp90.

To further validate this conclusion, we next investigated how LA1011 affects the binding of intact FKBP51. We therefore formed a complex consisting of 30 μM Hsp90 and 1 mM LA1011 and measured the binding for intact FKBP51. Using a one-site binding fit for intact FKBP51 provided an affinity for binding of *K*d = 2.1 μM (Figure 3G), while a two-site fit provided affinities of *K*d = 5.2 and *K*d = 2.7 μM (Figure 3H). Neither fit could reproduce the high-affinity binding of intact FKBP51 to Hsp90 (*K*d = 0.5 μM; Figure 3B) previously seen, suggesting that the binding of intact FKBP51 to the extreme hydrophobic pocket of Hsp90 had been compromised. The results, collectively, show that only one molecule of intact FKBP51 simultaneously binds to the MEEVD motif and the hydrophobic pocket at the extreme C-terminal dimer interface of Hsp90 and that binding of a second molecule is limited to just the MEEVD motif in our in vitro system. However, when the hydrophobic pocket of Hsp90 is occupied by LA1011, two FKBP51 molecules are limited to simultaneously binding to just the MEEVD motifs of Hsp90.

### 3.3. Effects of LA1011 on the Binding of Other TPR-Domain-Containing Co-Chaperones of Hsp90

The binding of LA1011 to the extreme C-terminal hydrophobic pocket of Hsp90 clearly disrupts the binding of FKBP51. However, other TPR-domain-containing co-chaperones also contain an equivalent extended helix 7 of their TPR domains, which may also bind to the C-terminal hydrophobic pocket of Hsp90. The most important residues of the helical extension of FKBP51 involved in binding to Hsp90 were Ile 408, Tyr 409, Met 412, Phe 413, Phe 416 and Ala 417 (IY--F--FA; Figure 4A,B). FKBP52 had a similar conserved motif (LY--MF--LA), where Ile 408 and Phe 416 of FKBP51 were replaced by Leu 409 and Leu 417 of FKBP52, respectively (Figure 4B). The degree of conservation suggests that both co-chaperones would bind in a similar manner to the extreme C-terminal hydrophobic pocket of Hsp90. Cyp40, however, appeared to have what looks like a truncated motif (VY--MF-), where FKBP51 Ile 408 is replaced by Val 364 and the downstream residues, equivalent to Phe 416 and Ala 417 of FKBP51, were missing altogether in Cyp40 (Figure 4B). Another example of a TPR helix extension binding to the hydrophobic pocket of Hsp90 was recently seen in two EM structures of Hsp90 in complex with PP5 (serine/threonine–protein phosphatase), CDC37 (cell division cycle 37) and kinase [49,50]. Significantly, it was reported that Phe 148 and Ile 152 of human PP5 are bound in the extreme C-terminal hydrophobic pocket of Hsp90 [50], but our structural comparisons showed that helix 7 of the PP5 TPR-domain adopts a completely different conformation relative to that of FKBP51 (Figure 4C). The TPR helical extension of human PP5 also possesses two conserved acidic residues, Asp 155 and Glu 156, that are in close proximity to the basic residues, Arg 679 and Arg 682, of Hsp90 [50]. In yeast PPT1, these residues are also conserved (Phe 132 (PP5-Phe 148), Ile 136 (PP5-Ile 152) and Glu 140 (PP5-Glu156)), except for Asp 155 of human PP5, which is an alanine residue instead (Ala 139). Although the binding conformations of the helical extension of PP5 and FKBP51 are different, nonetheless, the residue positions equivalent to Tyr 409, Met 412 and Phe 413 of FKBP51 appear to be conserved and therefore represent the most important residues for the interactions seen to date (Figure 4B). It is also likely the binding of the helical extension of PP5 would be compromised by Hsp90-bound LA1011 (Figure 4D). In another study, non-sense mutations of the TPR helical extension of aryl hydrocarbon receptor-interacting protein (AIP), which cause termination at Arg 304 and Gln 307, were found to affect client protein maturation [43]. Consequently, the mechanism of client maturation by AIP might be disrupted because the helical extension of AIP fails to dock with the extreme C-terminal hydrophobic pocket of Hsp90 (Figure 4E). Thus, failure to correctly dock would presumably compromise client protein maturation. We noted that human AIP has three conserved residues (Phe 324, Ile 327 and Phe 328), which align with the conserved Phe 148, Ala 151 and Ile 152 residues of PP5 (Figure 4B) or Tyr 409, Met 412 and Phe 413 of FKBP51, respectively. In fact, a recent electron microscopy structure of AIP and Hsp90 shows that helix 7 of the TPR domain of AIP binds to the C-terminal hydrophobic pocket of Hsp90 in a similar manner to that seen for FKBP51 (Figure 4F). Lastly, we also identified a conserved motif in the TPR extension of CHIP consisting of Leu 161, Tyr 164 and Leu 165, which correspond to Tyr 409, Met 412 and Phe 413 of FKBP51, respectively. This is significant since CHIP is involved in the degradation of abnormal or modified tau [20,21] and disruption of the CHIP–Hsp90 complex could increase tau aggregation. However, it would appear from our mice models that this may not be so significant.

This brief analysis suggests that the docking conformation of the helical extension of TPR-domain-containing co-chaperones of Hsp90 might be variable, or at least there are two conformational binding states that we have so far observed: one seen with PP5 and the other with FKBP51 and AIP. Although the two binding conformations are different, they are dependent on a set of conserved residues, namely the equivalent residue positions Tyr 409, Met 412 and Phe 413 of FKBP51. In both these states, a key phenylalanine residue is well placed in the binding position of the trifluoromethyl phenyl group of LA1011 (Figure 4D). It would appear that the effects of LA1011 on the function of TPR-domain-containing co-chaperones of Hsp90 would be limited to those that dock with the C-terminal hydrophobic pocket of Hsp90. However, whether LA1011 affects the dynamic nature of the Hsp90 chaperone cycle, and therefore the function of a more diverse group of Hsp90 co-chaperones, still remains to be explored. Clearly, further work is required to establish a clearer picture.

## 4. Discussion

The structure presented herein showed that LA1011 binds to the opposite end of the C-terminal domain of Hsp90 in relation to the site suggested using molecular dynamics. However, some caution should be exercised in completely disregarding the molecular dynamics site, as both sites could be targets of LA1011. Thus, it is possible that our crystal structure is selective for LA1011 binding to only the extreme C-terminal hydrophobic pocket. We in fact note that the binding of LA1011 to an Hsp90-FKBP51 complex does not in itself completely block LA1011 binding. This could be due to competition between FKBP51 and LA1011, or alternatively, binding of LA1011 could be occurring at more than one site, such as the molecular dynamics site identified previously. It is clear that further work is required in order to clarify the exact situation. However, this does not escape the fact that it is clear that LA1011 directly competes with FKBP51 for binding to the extreme C-terminal hydrophobic pocket of Hsp90. It appears that the trifluoromethyl phenyl group of LA1011, to some degree, mimics a key binding residue of FKBP51 (Phe 413), PP5 (Phe 148) and AIP (Phe 324), such that future development of LA1011 would need to focus on this key interaction. Thus, the binding interactions seen for FKBP51, PP5 and AIP will now form the basis for the design of improved small molecules that can interact with the Hsp90 hydrophobic pocket, which we are now actively pursuing.

Previous studies suggest that FKBP51 increases with age and stress and increases even further with AD disease, and it significantly increases the hyperphosphorylation of tau [25,27]. Since FKBP51 is thought to favour tau fibrillation, it would be reasonable to assume that decreasing the association of FKBP51 with tau-bound Hsp90 would decrease the ability of this system to promote the hyperphosphorylation of tau [11,12]. We therefore suggest that our previous observations where LA1011 increased dendritic spine density and reduced tau pathology and amyloid plaque formation in transgenic AD mice could be directly due to the disruption of the Hsp90-FKBP51 complex [41]. Thus, in vivo, where the cellular concentration of the Hsp90-FKBP51 complex is ‘abnormally’ elevated (due to high levels of FKBP51), LA1011 might restore ‘normal’ cellular levels of the Hsp90-FKBP51 complex by reducing the binding of FKBP51 to Hsp90. This in turn could then reduce the phosphorylation of tau and might therefore limit disease progression. However, we currently do not have direct data to support this hypothesis so consequently, further biological studies using cells and a mouse model for AD are in progress to gain such evidence in support of our concepts described herein. Thus, at this stage, we are unable to completely exclude whether disruption of other immunophilin–Hsp90 complexes by Hsp90 or specific protein clients of Hsp90 might also play a role in improving the prognosis of AD, as seen in our mouse models.

The fact that numerous TPR-domain-containing co-chaperones of Hsp90 are likely to be affected by LA1011, it follows that numerous Hsp90 processes would also be affected by targeting the extreme C-terminal hydrophobic pocket of Hsp90. Nonetheless, mouse model studies suggest a positive prognosis with LA1011 treatment [41], and consequently, this makes it a target of prime importance for further studies, especially as drugs to treat AD have been difficult to develop due to a lack of understanding of the underlying mechanism that leads to this disease [51]. Here, we presented a possible mechanism that appears to influence AD progression that could be targeted with a small molecule to help to improve the prognosis of AD. Since AD is generally a disease that affects the elderly, small changes in the progression of such a disease could have profound effects on the quality of life for such patients.

## Figures and Tables

**Figure 1 biomolecules-13-01051-f001:**
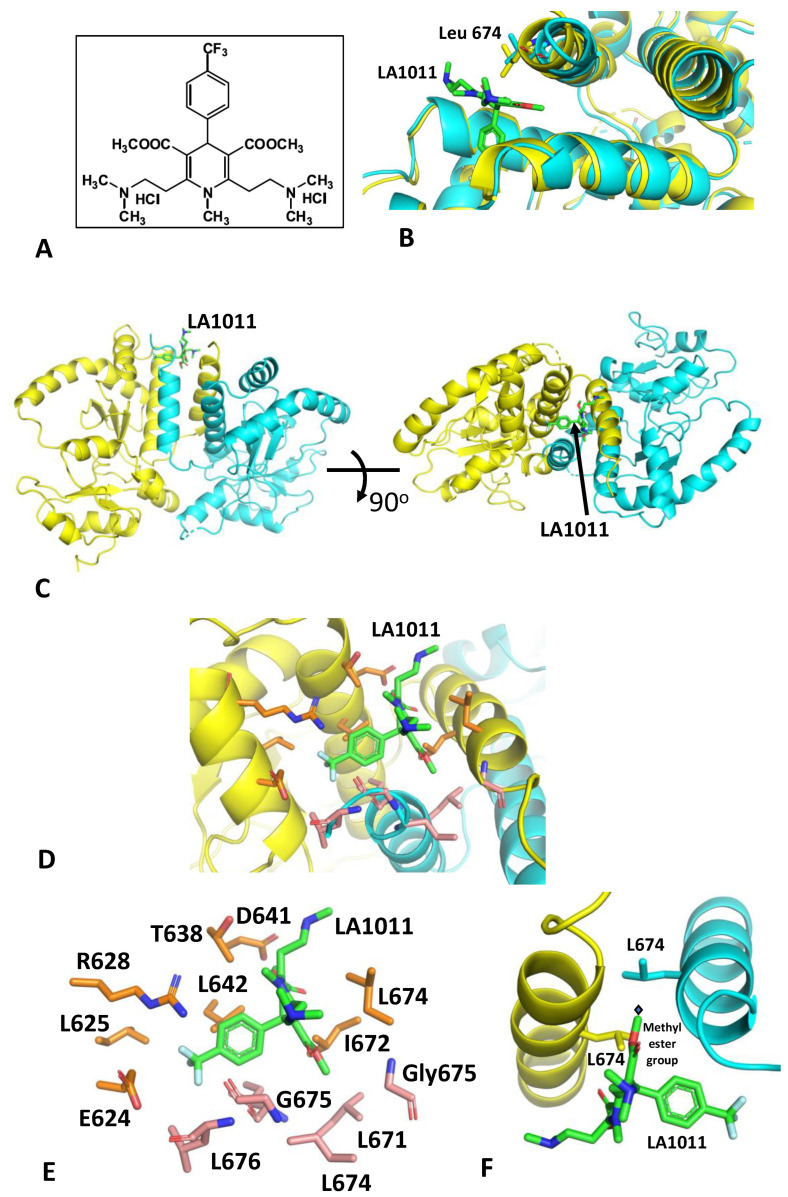
The structure of the Hsp90-LA1011 complex. (**A**) Structure of LA1011. (**B**) PyMol cartoon showing the superimposition of the Hsp90 dimer occupied with LA1011 (cyan) and the apo-dimer (yellow) seen in the crystals. It is evident that the LA1011-binding site in the apo-dimer is particularly restricted by the side chain of Leu 674. Binding would require a significant movement of the helix represented by residues Pro 661 to Leu 676. (**C**) PyMol cartoon showing orthogonal views of the Hsp90-LA1011 complex. The two protomers of the C-terminal domain of the Hsp90 dimer are shown in yellow and cyan. LA1011 is shown in green stick format. (**D**) PyMol cartoon showing a close-up of the LA1011-binding site of Hsp90. The two protomers of the C-terminal domain of the Hsp90 dimer are shown in yellow and cyan and amino acid side chains in gold and salmon stick format, respectively. LA1011 is shown in green stick format. (**E**) PyMol cartoon showing the side chains of Hsp90 (gold sticks form protomer A and salmon sticks form protomer B) that form the hydrophobic pocket of Hsp90 to which LA1011 (green sticks) is bound. (**F**) PyMol cartoon showing two symmetrically related helices (helix shown is Pro 661 to Ser 673) of the Hsp90 C-terminus. The symmetrically related Leu 671 residues are shown in stick format. The symmetry point is shown by the black diamond. The methyl ester group of LA011 is close to the symmetry point, thus restricting the binding of a second LA1011 molecule to the C-terminal domain of Hsp90.

**Figure 2 biomolecules-13-01051-f002:**
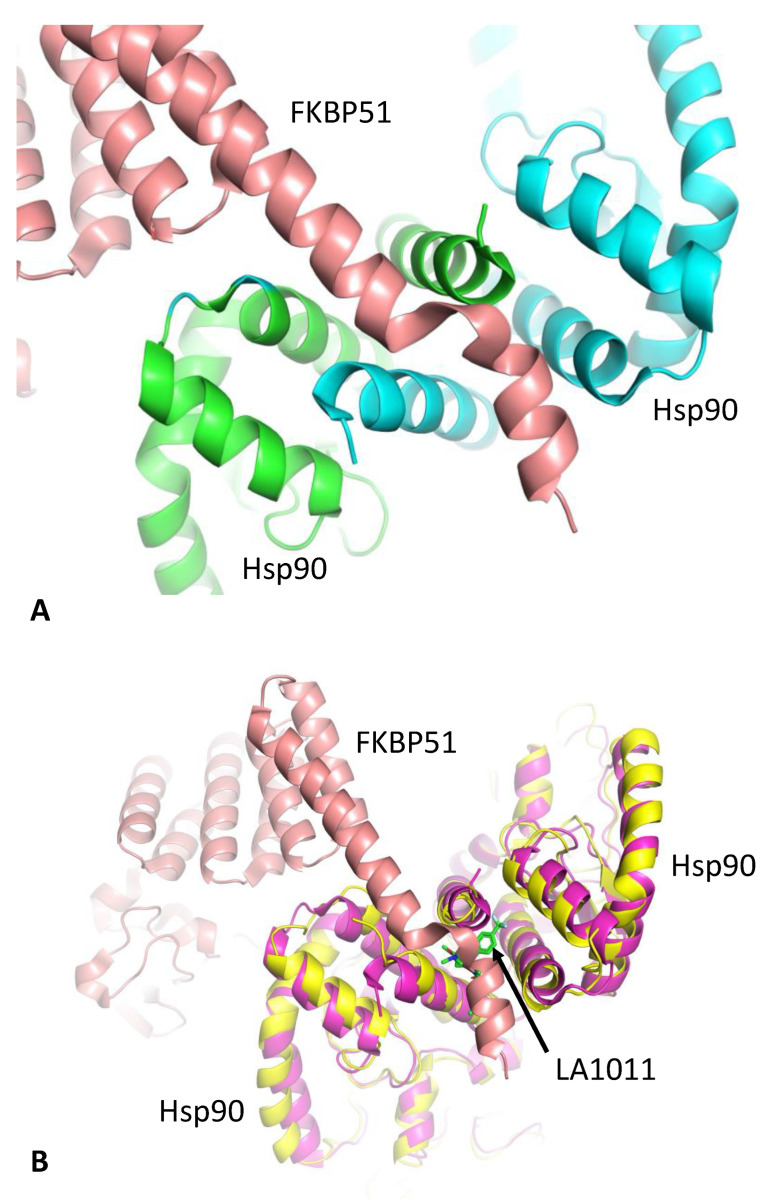
The structure of the Hsp90-FKBP complex. (**A**) PyMol cartoon showing helix 7 (salmon) of the FKBP51 TPR domain binding to the C-terminal domains of Hsp90 (green and cyan). (**B**) Superimposition of the Hsp90-FKBP51 complex (magenta and salmon, respectively) with the Hsp90-LA1011 complex (yellow and green sticks, respectively). Competition for binding between LA1011 and helix 7 of FKBP51 to the extreme C-terminal hydrophobic pocket of Hsp90 is evident through steric clashes with LA1011.

**Figure 3 biomolecules-13-01051-f003:**
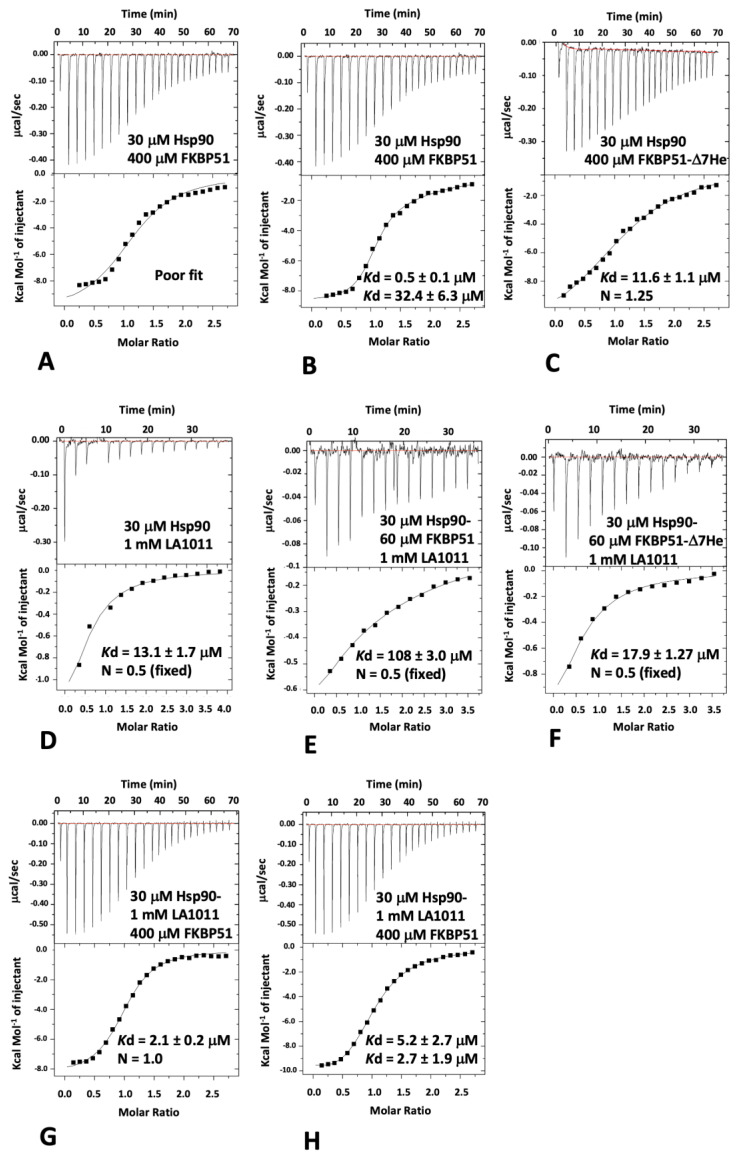
Isothermal titration calorimetry binding-studies for FKBP51 and LA1011 with Hsp90. (**A**) Titration of FKBP51 into Hsp90 and fitted with a one-site binding model that does not fit well to the experimental data. For panels A to H, the upper panel of each graph represents the raw data of the interaction experiment. The lower panel represents the fitted data. (**B**) Titration of FKBP51 into Hsp90 and fitted as a two-site binding model that fits the experimental data better than the one-site binding model (**A**). (**C**) Binding of the FKBP51-Δ7He construct to Hsp90, showing that the experimental data can be fitted with a one-site binding model. (**D**) Binding of LA1011 to Hsp90. (**E**) Binding of LA1011 to the Hsp90-FKBP51 complex and (**F**) to the Hsp90-FKBP51-Δ7He complex, showing that LA1011 binding is only compromised with the Hsp90-FKBP51 full-length complex. (**G**) Binding of FKBP51 to the Hsp90-LA1011 complex fitted as a one-site binding model and (**H**) fitted as a two-site binding model. The affinities in both fittings for binding of FKBP51 are similar and do not show the high-affinity binding observed in Figure 3B (*K*d = 0.5 μM), suggesting that FKBP51 only binds to the MEEVD motif of Hsp90 in the presence of LA1011.

**Figure 4 biomolecules-13-01051-f004:**
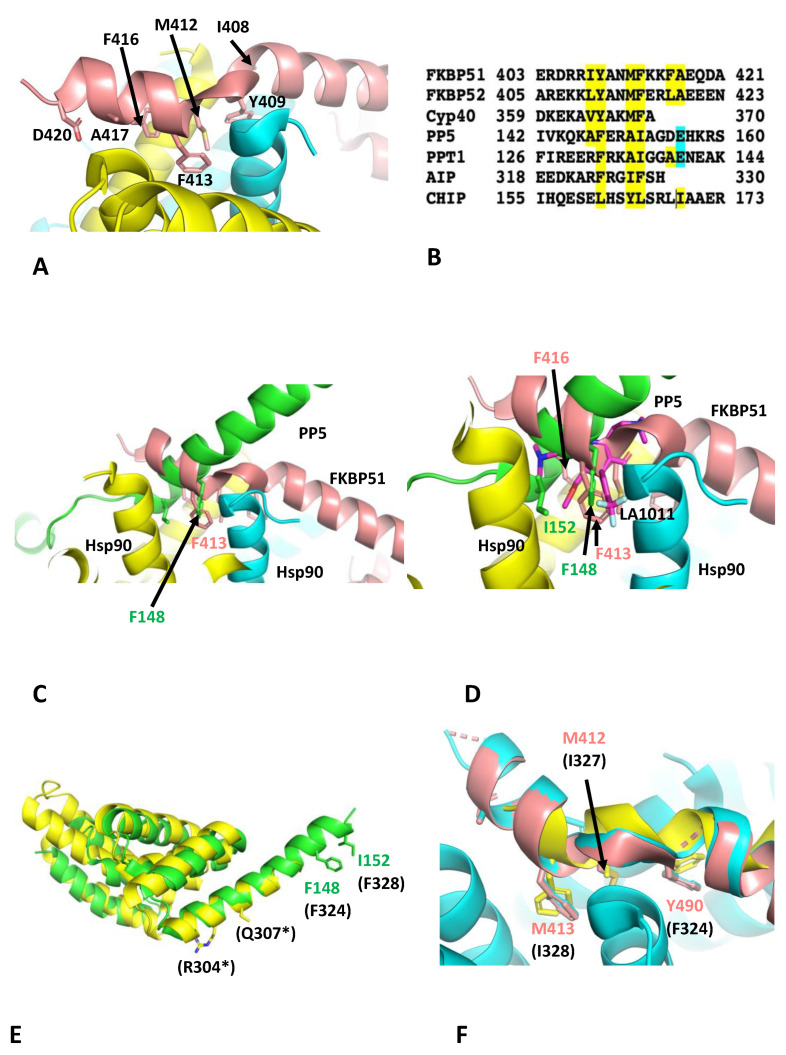
TPR domain binding to the extreme C-terminal hydrophobic pocket of Hsp90. (**A**) PyMol cartoon showing the critical residues (stick format) of FKBP51 (salmon) that interact with the hydrophobic pocket of Hsp90 (yellow and cyan). (**B**) Sequence alignment of segments of the helix extension of selected TPR-domain-containing co-chaperones of Hsp90. Amino acid numbering is shown. Yellow highlight, conserved residue positions relative to the FKBP51 motif involved in Hsp90 binding; cyan, the glutamate residues (Glu 156 of PP5 and Glu 140 of PPT1) of the protein phosphatases involved in binding to the hydrophobic pocket of Hsp90. Overall, the most significant residues for binding to Hsp90 are positions equivalent to Tyr 409, Met 412 and Phe 413 of FKBP51. (**C**) PyMol cartoon comparing the conformation of the bound TPR helix extension of FKBP51 (salmon) and that of PP5 (green) to the hydrophobic pocket of Hsp90 (cyan and yellow). (**D**) PyMol cartoon showing that LA1011 (magenta sticks) sterically clashes with the TPR helix extension of PP5 (green) and FKBP51 (salmon). Yellow and cyan are Hsp90. (**E**) PyMol cartoon showing the superimposition of PP5 (green) and AIP (yellow). Truncation of AIP at Arg 304 and Gln 307 would disrupt binding to the C-terminal hydrophobic pocket of Hsp90, since the downstream residues Phe 324 (F148 in PP5) and Phe 328 (Ile 152 in PP5) required for this interaction would be lost. Residue numbers are shown for PP5, and those in parentheses are for AIP. (**F**) PyMol cartoon showing the superimposition of the helical extension of FKBP51 (salmon) and AIP (yellow) involved in binding to Hsp90 (cyan). Residues for FKBP51 are shown, and those in parentheses are for AIP.

**Table 1 biomolecules-13-01051-t001:** Crystallographic statistics.

Data Set (Highest Shell in Parentheses)	AutoPROC	STARANISO
a (Å)	97.71	
b (Å)	97.71	
c (Å)	275.57	
α (°)	90.0	
β (°)	90.0	
γ (°)	90.0	
Space group	P 4_1_ 2_1_ 2_1_	
Wavelength (Å)	0.96864	
Resolution limit (Å)	92.1–3.18 (3.23–3.18)	92.1–2.94 (3.19–2.94)
Number of unique obs.	22,841 (1103)	23,054 (1153)
Completeness (%)	97.4 (97.0)	91.9 (58.1)
Multiplicity	9.3 (5.9)	9.2 (4.0)
Rmerge (I) %	0.218 (1.867)	0.217 (1.215)
Rpim (I) %	0.073 (0.803)	0.073 (0.666)
CC_1/2_	0.995 (0.408)	0.995 (0.361)
I/σI	9.0 (1.1)	9.0 (1.4)
Refinement		
Resolution range (Å)		79.7–2.94
Rcryst		0.2470
Rfree		0.3062
Number of protein atoms		6816
Number of ligand atoms		0
Number of solvent atoms		1
Mean B		70.11
Rmsd bond lengths (Å)		0.005
Rmsd bond angles (°)		1.051

## Data Availability

Raw data for ITC experiments can be found at https://doi.org/10.25377/sussex.22341712. The structure was deposited in the PDB database (PDB 8OXU). PDB files used in structural comparisons include AIP, PDB 7ZUB and PP5, 8GAE.

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
