# Peer review of "The Crystal Structure of the Hsp90-LA1011 Complex and the Mechanism by Which LA1011 May Improve the Prognosis of Alzheimer’s Disease"

_biomolecules, 2023, doi:10.3390/biom13071051_

Round 1
Reviewer 1 Report
In his study, Roe et al., determined the crystal structure of the small molecule dihydropyridine, LA1011, bound to the Hsp90 CTD fragment dimer. Previously this group identified LA1011 as a small molecule that reduces AD tau/amyloid pathology in a mouse model. Here, from their sturcture they identify interaction mode of LA1011 with Hsp90. LA1011 binds to a hydrophobic pocket present at the C-terminal domain interface of Hsp90, which is the opposing site from the previous molecular dynamics prediction. The binding mode of LA1011 was verified using FKBP51 which binds to the same pocket. LA1011 and FKBP51 competitively bind to the pocket and deletion of a helix of FKBP51 diminishes the competition which agree well with the structure and previous study. The structure provides a basis of not only understanding mechanism of action of LA1011, but also for aiding the design improvements of LA1011 with higher specificity and efficacy. These are interesting results and an important contribution to the molecular chaperone and aging fields. The experimental data are of high quality. There are only minor concerns/questions to be considered for revision.
One concern is that the role of Hsp90 in AD and tau accumulation is not nearly as well as established as generally indicated in the manuscript. For example, line 30: “the role of Hsp90 in Alzheimer’s Disease is well documented” cites a single, fairly limited review that focuses on extracellular chaperones in Abeta clearance and tau clearance Hsp90/Hsp70-CHIP. These processes are not well-understood and somewhat controversial in terms of effects on AD progression. The authors should soften their phrasing in the abstract and main text regarding Hsp90 and AD.
The authors focus on LA1011 in the disruption of FKBP51 binding as the mechanism of action. However, there are a number of immunophilin co-chaperones that likely bind this CTD groove and disrupting any or all of these co-chaperone interactions may result in the in vivo affects described previously. The authors should dampen their conclusion (statements in the Introduction and Discussion) that it is strictly disruption of the Hsp90-FKBP51 interaction that results in mouse phenotype. Additionally the affects in tau/Abeta could be more indirect such as a loss of nuclear receptor function since these are clients of Hsp90-immunophilins.
Alternatively it would be very interesting to compare binding affects across a number of related immunophilin co-chaperones to test whether their interactions are similarly inhibited in the presence of LA1011.
Why is binding sub-stoichiometric in the crystal? It is unclear why there is an unbound dimer in the asymmetric unit given the relatively good affinity of LA1011.
In Fig. 3 there are protein concentration typos and the black/red traces are not explained in the legend or fig.
The cryo-EM structures of AIP (XAP2) bound to Hsp90 are available now. These structures should be included as well to be compared to FKBP51 and PP5.
Reviewer 2 Report
Title: The Crystal Structure of the Hsp90-LA1011 Complex and the Mechanism by which LA1011 may Improve the Prognosis of Alzheimer’s Disease.
This manuscript reported the crystal structure of a C-terminal domain of Hsp90 in complex with LA1011, a chemical that was shown to reduce tau pathology and amyloid plaque formation in transgenic AD mice. It was proposed that the helical extension of the TPR domain of FKBP51 and LA1011 compete for binding to the extreme C-terminal hydrophobic pocket of Hsp90. However, this manuscript is not publishable in Biomolecules.
My comments are shown below:
1. Lines 23-24, the data presented in this work supported that LA1011 disrupts the interactions between Hsp90 and FKBP51, but more data is needed to demonstrate that it helps rebalance the Hsp90-FKBP51 chaperone machinery and provides a favorable prognosis toward AD.
2. Lines 84-87, this study solved the co-crystal structure of LA1011 in a complex with a fragment of Hsp90. No data is provided to show that their binding may help to restore normal levels of the FKBP51-Hsp90 complex in cells. Here has the same problem as point 1. These statements are hypotheses or possible explanations of the experimental data. The authors should not give too much speculation about their experimental results.
3. Lines 87-89, is the chemical LA1011 used to treat AD patients? This study did not do clinical research on the chemical LA1011. The clinical significance of this study may be overestimated.
4. The figure titles in figures should be removed.
5. For Figure 3, the ITC data should be better presented. The unit of protein concentration is not shown correctly. What are the red symbols and black symbols? The Kd and N numbers can be listed in a separate table.
6. The Discussion and Introduction give similar information. The authors should re-write one of these two sections.
Reviewer 3 Report
I have read with great interest the article by Roe et al. I find the study very interesting, with a lot of potential for future directions in the pharmacological treatment of proteinopathies, such as AD. My only comment is regarding the potential limitations that arise by the use of the C-terminal fragment of Hsp90 and not the complete full length protein. I think that this is an issue that should be at least discussed in the final section or/and explain why it is not possible to use the full length.
Otherwise, I endorse the publication of the study.
Reviewer 4 Report
Comments attached in separate file.

Round 2
Reviewer 2 Report
The quality of this manuscript is improved after revision.
In Figure 3, the protein concentration unit should be uM. It doesn't show properly in this figure.
Author Response
Thank you for your time in reviewing the manuscript and for your useful suggestions. On our version of the manuscript we can clearly see uM, using the greek letter mu. For the LA1011 1 mM is correct. I have attached a pdf of the powerpoint version that may allow you to se this, as I believe this may be a display issue at your end. Consequently, we cannot see any errors in figure 3 and no further amendments are required.

Reviewer 4 Report
The authors have addressed the major concerns in the manuscript raised by this reviewer. The manuscript can be accepted for publication.
There is a typo error in Figure 1 legend that needs to be corrected. Lue 674, Lue 671 needs to be changed to Leu 674 and Leu 671.
Author Response
Thank you for your time and useful suggestions. We have corrected Lue to Leu as requested.